# The Impact of Statins on Disease Severity and Quality of Life in Patients with Psoriasis: A Systematic Review and Meta-Analysis

**DOI:** 10.3390/healthcare12151526

**Published:** 2024-07-31

**Authors:** Abdulsalam Mohammed Aleid, Ghadah Almutairi, Rudhab Alrizqi, Houriah Yasir Nukaly, Jomanah Jamal Alkhanani, Deemah Salem AlHuraish, Hawazin Yasser Alshanti, Yaser Sami Algaidi, Hanan Alyami, Awatif Alrasheeday, Bushra Alshammari, Kawthar Alsaleh, Abbas Al Mutair

**Affiliations:** 1Department of Surgery, Medical College, King Faisal University, Hofuf 31982, Saudi Arabia; aaleid4@moh.gov.sa; 2Qassim College of Medicine, Qassim University, Buraydah 51452, Saudi Arabia; 391202826@qu.edu.sa; 3Department of Medicine and Surgery, Al-Qunfudah Medical College, Umm Al-Qura University, Makkah 24382, Saudi Arabia; s441010261@st.uqu.edu.sa; 4College of Medicine and Surgery, Batterjee Medical College, Jeddah 21442, Saudi Arabia; 130025.houriah@bmc.edu.sa; 5College of Medicine, Alrayan Medical College, Madina 42541, Saudi Arabia; 18120107@alrayancolleges.edu.sa; 6College of Medicine, Imam Abdulrahman bin Faisal University, Dammam 34212, Saudi Arabia; 2190005496@iau.edu.sa; 7College of Medicine, King Abdulaziz University, Jeddah 21589, Saudi Arabia; halshanti0001@stu.kau.edu.sa; 8School of Medicine, Royal College of Surgeons in Ireland, D02 YN77 Dublin, Ireland; yaseralgaidi20@rcsi.com; 9Department of Medical and Surgical Nursing, College of Nursing, Princess Norah Bint Abdurrahman University, Riyadh 11564, Saudi Arabia; hmalyami@pnu.edu.sa (H.A.); abbas.almutair@almoosahospital.com.sa (A.A.M.); 10Nursing Administration Department, College of Nursing, University of Hail, Hail 21424, Saudi Arabia; a.alrasheeday@uoh.edu.sa; 11Medical Surgical Nursing Department, College of Nursing, University of Hail, Hail 2440, Saudi Arabia; bu.alshammari@uoh.edu.sa; 12Research Center, Almoosa Specialist Hospital, Al-Ahsa 36342, Saudi Arabia; 13Almoosa College of Health Sciences, Al-Ahsa 36342, Saudi Arabia; 14School of Nursing, University of Wollongong, Wollongong, NSW 2522, Australia; 15Department of Nursing, Prince Sultan Military College of Health Sciences, Dhahran 31932, Saudi Arabia

**Keywords:** psoriasis, statins, severity, quality of life

## Abstract

Background: Psoriasis, a chronic autoimmune condition, imposes significant burdens on patients’ well-being. While corticosteroid medications are commonly used, their prolonged use presents risks. Statins, known for their immunoregulatory and anti-inflammatory properties, have emerged as potential alternatives. Previous reviews indicated that statins might improve psoriasis symptoms but showed inconsistent results and lacked meta-analyses that generated pooled effect estimates. Therefore, this study addresses this gap by providing a comprehensive overview of the impact of statins on psoriasis severity and quality of life (QoL) for patients with psoriasis. Methods: A thorough search of four electronic databases (PubMed, Cochrane Central Register of Controlled Trials, Scopus, and Science Direct) was conducted for relevant studies published before April 2024. Results: Seven studies involving 369 patients were included. This meta-analysis showed a statistically significant reduction in PASI scores at week 8 with statin treatment (MD = −1.96, 95% CI [−3.14, −0.77], *p* = 0.001). However, no statistically significant difference was found between statins and placebo at week 12 (MD = 0.19, 95% CI [−0.18, 0.55]). Additionally, DLQI scores indicated a significant improvement in quality of life with statins compared to placebo (MD = −3.16, 95% CI [−5.55, −0.77]). Conclusions: Statins can improve disease severity and quality of life in psoriasis patients, suggesting the potential benefits of statin therapy. However, further research is needed to determine the optimal treatment duration, address outcome heterogeneity, and explore additional benefits such as cholesterol and triglyceride reduction.

## 1. Introduction

Psoriasis is a chronic immune-mediated dermatological condition of genetic origin that primarily affects the skin and joints. Psoriasis impacts around 125 million people globally with new cases occurring at a rate of approximately 80 per 100,000 person-years [1,2]. The prevalence of psoriasis generally increases gradually from about 0.12% at 1 year of age to 1.2% by age 18. While corticosteroids are commonly used to manage psoriasis, their prolonged use poses risks, prompting an investigation into alternatives, including statins, which may alleviate psoriasis symptoms by possessing immunoregulatory and anti-inflammatory properties [3].

The pathogenesis of psoriasis mainly involves an autoimmune response, which is triggered by the activation of inflammatory cells via a series of cytokine activations, with the IL-23-mediated activation of the Th17 pathway playing a central role. This activation leads to subsequent events, including the proliferation of keratinocytes and increased expression of angiogenic mediators and endothelial adhesion molecules. This process enhances the migration of immune cells into the skin, resulting in the inflammation and lesions characteristic of the disease [3,4]. Factors such as genetic predisposition, aging, climate, sun exposure, and ethnic background also contribute to the likelihood of developing psoriasis [5]. Psoriasis is characterized by three main clinical features: thickening, erythema, and scaling. The disease manifests in several subtypes, including plaque psoriasis (or psoriasis vulgaris). Plaque psoriasis is the most common form, representing approximately 90% of all cases, and is characterized by monomorphic, well-defined erythematous plaques with silver lamellar scales. These plaques may cover large areas of the body or, in severe cases, develop into erythroderma that affects the entire body. Scalp involvement is common, occurring in 75–90% of cases. Other subtypes include guttate psoriasis, which is marked by scaly teardrop-shaped spots; eruptive psoriasis, similar to guttate with sudden onset of scaly lesions; inverse psoriasis, typically appearing in skin folds and also known as intertriginous or flexural psoriasis; pustular psoriasis, which is divided into localized forms affecting palms and soles or a generalized type; and erythrodermic psoriasis, a rare but severe manifestation [6,7,8].

The treatment of psoriasis varies widely, depending on the severity and localization of the condition. Mild or localized forms can often be managed with corticosteroids, vitamin D analogs, calcineurin inhibitors, and targeted phototherapy. Corticosteroids are fundamental in treating mild or localized psoriasis, primarily through their anti-inflammatory and antiproliferative effects, which suppress pro-inflammatory cytokines [9,10]. However, extended use over large areas can pose risks to the pituitary or adrenal axis. Vitamin D analogs inhibit the differentiation and proliferation of keratinocytes by blocking vitamin D receptors. Studies have shown that combining vitamin D analogs with topical corticosteroids is particularly effective [11]. For cases with moderate to severe psoriasis, systemic treatments may be required and can include biologics and oral agents like methotrexate or cyclosporine, often used alongside topical treatments or phototherapy [12]. In addition, biologics are among the latest advancements in psoriasis treatment. These medications work by targeting specific cytokine molecules in the inflammatory cascade, including TNF-α, IL-17, or IL12/23 inhibitors [13].

Statins are lipid-lowering drugs are commonly used in patients with hypercholesterolemia due to their mechanism of action in inhibiting 2-hydroxyl-methyl-glutaryl coenzyme A (HMG-CoA) reductase, which has an important role in cholesterol biosynthesis [14]. Statins are primarily used for adult patients with a greater risk of developing cardiovascular diseases, as they play a significant role in lowering the risk of all-cause mortality and cardiovascular events [15].

However, in recent years, statins have received significant attention for the compelling evidence of their role in immunomodulation, by which they provide therapeutic benefits in managing autoimmune-mediated disease. According to several studies, statins can reduce inflammation and affect immune responses through both mevalonate pathway-dependent and -independent mechanisms, making them a subject of interest in the management of autoimmune disorders including rheumatoid arthritis, psoriasis, multiple sclerosis, and systemic lupus [16,17,18,19,20].

The use of statins for patients with psoriasis is becoming more common due to their typically abnormal lipid profiles, including high low-density lipoprotein (LDL), triglycerides (TGs), and total cholesterol (TC) and low high-density lipoprotein (LDL) [21]. Nevertheless, it is found that statin improves psoriasis symptoms due to its immunomodulatory and anti-inflammatory effect through the inhibition of leukocyte function antigen-1 (LFA-1), interleukin-1 and -6, and tumor necrotic factor alpha (TNF-α), and its ability to reduce CRP levels. It has an antiproliferative effect by inhibiting vascular endothelial growth factor (VEGF), which induces vascular proliferation, which is important in psoriasis pathophysiology [22,23]. This explains the rationale for using statins in the treatment of psoriasis.

A previous systematic review included three clinical trials evaluating the efficacy of statins on the severity of psoriasis, reporting inconsistent outcomes [24]. Later, another systematic review was conducted on various lipid-lowering agents, including RCTs, single-arm studies, and in vitro studies, finding that cholesterol-reducing medications, particularly statins, may alleviate symptoms in psoriasis patients with promising efficacy and minimal side effects [25]. However, neither of these studies conducted a meta-analysis that generated a pooled effect estimate. Therefore, this study aims to provide a comprehensive overview of the impact of statins on area and severity indices and quality of life (QoL).

## 2. Materials and Methods

The process of conducting this systematic review and meta-analysis adhered to the standards set by the Preferred Reporting Items for Systematic Reviews and Meta-Analyses (PRISMA) and the Cochrane Handbook for Systematic Reviews of Interventions [26,27]. Moreover, the study was officially registered with PROSPERO under the given registration number (CRD42023479379).

### 2.1. Database Searching

Four electronic databases (PubMed, Cochrane Central Register of Controlled Trials, Scopus and Science Direct) were searched for articles investigating the impact of statins in area and severity indices and their effect on QoL for patients with psoriasis. All articles published before April 2024 were included. A mix of search terms was utilized for this purpose. We utilized the following keywords in the search strategy and combinations using Booleans AND/OR: “Statin”, “hydroxymethylglutaryl-CoA reductase inhibitors”, “Atorvastatin”, “Simvastatin”, “Rosuvastatin, “psoriasis”, “psoriatic lesion”, and “plaque psoriasis” without time restrictions.

### 2.2. Inclusion and Exclusion Criteria

#### 2.2.1. Inclusion Criteria

For quantitative assessment, this study included all RCTs comparing statins against placebo in patients aged 16 years or older fulfilling the clinical and morphological criteria of mild, moderate, or severe plaque psoriasis, which is confirmed clinically by a dermatologist before participating in studies, and those with a previous history of confirmed psoriasis or psoriatic medications and RCTs measuring severity and QoL indices.

#### 2.2.2. Exclusion Criteria

Articles with designs other than RCTs, such as reviews, letters to editors, abstracts, opinions, and non-human studies, were excluded from the meta-analysis. Trials that used drugs other than statins, additional interventional drugs, or other comparators instead of placebo or were not published in English were also excluded. Studies conducted on non-psoriatic or healthy participants were as well.

### 2.3. Study Selection and Data Extraction

Following the database search, results were imported into Endnote software version 20 for the purpose of identifying and removing duplicate entries [28]. Following this, Rayyan web-based software (https://www.rayyan.ai/) was utilized by two authors to independently review the titles and abstracts of all studies. The two authors evaluated the relevance of the studies [29]. In instances of disagreement, the reviewers held discussions to achieve consensus with a third reviewer to make the final decision. Articles meeting the inclusion criteria underwent full-text screening. Moreover, scrutinization of the reference lists of the included studies was performed to ensure no relevant studies were overlooked. Finally, data extraction from the selected articles, including publication year, target population, baseline characteristics, study locations, and outcomes, was performed manually by two independent reviewers, who recorded the data in a Google sheet. In cases of data extraction inconsistencies, the authors engaged in discussions to reach a consensus.

### 2.4. Primary Outcomes

This study focused on psoriasis severity and QoL as the primary outcomes. The severity of psoriasis was investigated using the Psoriasis Area Severity Index (PASI) after 4 weeks, 8 weeks, and 12 weeks. The PASI is designed to evaluate the extent of psoriasis and its severity based on the area affected. The PASI score is calculated by dividing the human body into four sections for assessment: head, upper extremities, trunk, and lower extremities, representing 10%, 20%, 30%, and 40% of the total body surface area (BSA), respectively. By utilizing a rating scale from 0 (none) to 4 (very severe), each of the body sections is evaluated independently for erythema, induration, and scaling. Six classification levels are used for the assessment of psoriatic extent including 0 (no involvement); 1 (1% to 9%); 2 (10% to 29%); 3 (30% to 49%); 4 (50% to 69%); 5 (70% to 89%); 6 (90% to 100%). Regarding the calculation of the PASI score, the following formula is used:PASI = 0.1 (Eh + Ih + Sh) Ah + 0.2 (Eu + lu + Su) Au + 0.3 (Et +lt + St) At + 0.4 (El +ll +Sl) Al 

In this formula, E denotes erythema, I denotes induration, and S denotes scaling. The variable A represents the area, while h, u, t, and l correspond to the scores for the head, upper extremities, trunk, and lower extremities, respectively [30,31].

For the assessment of QoL, the 10-item Dermatology Life Quality Index (DLQI) was used. This instrument examines QoL through assessing six different dimensions, including symptoms and feelings, leisure, daily activities, personal relationships, work and school performance, and treatment. Each item of the DLQI is rated on a scale from 0 (not at all) to 3 (very much). The total DLQI score is obtained by summing the scores of all items, with the maximum possible score being 30. Higher scores indicate a greater impact on the QoL of the patient. The scores of DLQI can be as classified as no effect (0 to 1 points); small effect (2 to 5 points); moderate effect (6 to 10 points); very large effect (11 to 20 points); extremely large effect (21 to 30 points) [32].

### 2.5. Secondary Outcomes

The secondary outcomes of interest for this study were the rates of patients achieving a 75% reduction in PASI score and lipid profile (including triglyceride and cholesterol).

### 2.6. Risk of Bias Assessment

With the utilization of the Cochrane Risk of Bias tool (ROB1) of interventional studies, the risk of bias was evaluated for the included RCTs. Assessment of risk of bias was performed in terms of selection bias and allocation concealment, blinding of patients and personnel, blinding of outcome assessors, missing outcome data, selective reporting of outcomes, and other sources of bias if present [33]. Two authors independently classified each domain as high, low, or unclear risk. In cases of rating discrepancies, they were resolved by a meticulous reading of the assessed studies and consensus with a third author.

### 2.7. Statistical Analysis

Using RevMan software version 5.4.1, a meta-analysis was carried out in the presence of at least two included studies with data available for assessed outcomes [34]. Data were pooled as Mean Difference (MD) with 95% confidence intervals (CIs) for continuous outcomes. For dichotomous outcome data, the frequency of events and the total number of patients were combined and represented as a risk ratio (RR) with a 95% CI. A *p*-value of < 0.05 for statistical significance was established. Rather than extracting final numbers, the change from baseline values was extracted. When no standard deviation was reported, *p*-values were used to compute an estimate. In addition, RevMan was used to calculate standard errors or SD in case of incomplete data. To yield a more conservative estimate of the pooled effect and provide more generalizable results, a random effect model (inverse variance) was adopted rather than a fixed effect model. In accordance with chapter nine of the Cochrane Handbook, Chi-square and I-square tests were used to examine the presence and degree of heterogeneity, respectively. Concerning the interpretation of I-square test results, 0–40% denoted insignificant, 30–60% denoted moderate, and more than 50% denoted substantial. An alpha level less than 0.1 for the Chi-square test was considered significant for heterogeneity. Heterogeneity was solved by leaving out analysis (sensitivity analysis) or subgroup analysis according to the drug or end-point of assessment. Owing to the limited number of included studies, it was not feasible to evaluate publication bias using funnel plots.

## 3. Results

### 3.1. Study Selection

To gather potentially relevant records, a systematic literature search was conducted across various databases, including PubMed, Cochrane Central, Scopus, and Web of Science. EndNote was utilized to remove duplicate records, resulting in 21 duplicates being eliminated. Following this, 101 records underwent screening of titles and abstracts. From these, 11 articles were found to meet the eligibility criteria for the research question. In the final step, full-text assessments were conducted, leading to the exclusion of four records. Reasons for exclusion included two being conference abstracts, one having a different comparator, and one involving healthy participants instead of patients. Consequently, seven studies were incorporated into the qualitative and quantitative synthesis. These studies consisted of six clinical trials and one post hoc analysis [35,36,37,38,39,40,41]. The process of searching and the number of included and excluded studies are illustrated in Figure 1.

### 3.2. Characteristics of the Studies

All included RCTs evaluated the effectiveness of statins compared to placebo/vehicle in patients with psoriasis and were published between 2010 and 2024. The analysis included seven RCTs and one post hoc study, with a total of 369 patients [41]. Six studies were conducted in Asia, of which four were in Iran [35,37,39,40], one in Pakistan [38], and one in the Philippines [36], while only one study was conducted in the United Kingdom [41]. Three clinical trials assessed the effectiveness of oral atorvastatin [36,37,38]. Al Salman et al. [35] and Naseri et al. [40] assessed oral simvastatin, while Mohammadi et al. [39] assessed topical Rosuvastatin. Table 1 shows a detailed summary of the included studies. Table 2 illustrates the characteristics of patients in the included studies.

### 3.3. Risk of Bias and Quality Assessment

The quality of evidence was assessed, and the risk of bias was assessed for the interventional studies tool ROB1. The assessment of seven included trials revealed that the two studies by Al Salman et al. [35] and Chua et al. [36] were deemed high quality with low-risk judgment on all domains of ROB1. The three studies by Faghihi et al. [37], Jawed et al. [38], and Naseri et al. [40] were evaluated as unclear with some concerns due to one unclear judgment of detection bias on the study by Faghihi et al. [37]. Jawed et al. [38] showed unclear judgment on selection and detection biases due to insufficient information to judge. In addition, the study by Naseri et al. [40] was deemed unclear on selection and reporting biases. Furthermore, the study by Mohammadi et al. [39] was evaluated as low quality due to a high risk of bias on attrition bias resulting from unmanaged loss of patients on follow-up. The graph and summary for the risk of bias scores are illustrated in Figure 2 and Figure 3, respectively.

### 3.4. Primary Outcomes

#### 3.4.1. Psoriasis Area Severity Index (PASI)

A complete meta-analysis was conducted to evaluate the pooled effect of clinical trials for this outcome. The PASI score was measured at four, eight, and twelve weeks following the start of treatment. Two studies [36,39] revealed the PASI score at week 4; the findings showed no statistically significant difference between statins and placebo. PASI scores at week 4 were reported in two studies [36,39]. The results revealed no statistically significant difference between statins and placebo (pooled effect (MD = −0.60, 95% CI [−1.76, 0.56]) with a *p*-value of 0.31). Significant heterogeneity was noted (*p* = 0.01 and I2 = 85%). This inconsistency is attributed to different types of statins as patients in one study received topical Rosuvastatin 0.5% twice per day for 8 weeks, while in the other, they received atorvastatin 40 milligrams (mg) per day. The PASI score at week 8 was examined in four included trials [27,31,32,35]. The results of the meta-analysis indicated that there was a statistically significant reduction in the PASI score in the statin group compared to the placebo (MD = −1.96, 95% CI [−3.14, −0.77]), which had a *p*-value of 0.001. The results showed that no marked heterogeneity was detected (*p* = 0.11 and I2 = 50%). Moreover, assessment of PASI scores at week 12 after initiation was performed in four studies [35,36,37,38]. There was no observed statistically significant difference between statins and placebo at week 12 (MD = 0.19, 95% CI [−0.18, 0.55]) indicated by a *p*-value of 0.32 with no observed heterogeneity (*p* = 0.42, I2 = 79%). Figure 4 presents a forest plot of the meta-analysis for the PASI scores.

#### 3.4.2. Dermatology Life Quality Index (DLQI)

In this meticulous meta-analysis regarding DLQI outcome, detailed data results were extracted from four clinical trials [24,25,27,39]. The pooled effect revealed no statistically significant difference between statins and placebo (MD = −1.54, 95% CI [−4.60, 1.53]), indicated by a *p*-value of 0.33 as shown in Figure 5A; however, the data showed marked heterogeneity (*p* = 0.01 and I2 = 73%). This heterogeneity was solved by conducting a sensitivity analysis leaving out the study by Jawed et al. [38]. After the exclusion, the results showed a statically significant difference favouring statins over placebo (MD = −3.16, 95% CI [−5.55, −0.77]) with a p-value of 0.01 and heterogeneity Chi-square test p-value of 0.35 and the I2 test reduced to 6% as shown in Figure 5B.

### 3.5. Secondary Outcomes

#### 3.5.1. PASI 75% Reduction

Two studies reported a rate of patients achieving a 75% reduction in PASI score. The meta-analysis results revealed no statistically significant difference in rates between the two groups (RR = 1.05. 95% CI [0.55, 2.02]), confirmed by a *p*-value of 0.87. Data showed no heterogeneity among studies (*p* = 0.44 and I2 = 0%). Figure 6 shows the meta-analysis of this outcome in a forest plot.

#### 3.5.2. Lipid Profile

In this meta-analysis, a detailed extraction of lipid profile results from two trials was carried out [38,39]. The results showed a significant reduction in cholesterol and triglycerides, favoring the statin group (MD = 17.43, 95% CI [−26.31, −8.54] with a *p*-value of 0.001; MD = −6.47–5.95, 95% CI [−7.40, −5.53–4.93] with a *p*-value of <0.00001, respectively). However, on triglyceride outcome, 99.9% of the weight of results was derived alone, which influenced the robustness of these results. These findings indicate the positive effects of statins on lipid profile in psoriatic patients. Heterogeneity was deemed marked on cholesterol outcome (*p* = 0.08 and I2 = 67%), and no heterogeneity was detected on triglyceride outcome (*p* = 0.72 and I2 = 0%). Figure 7 shows the meta-analysis of lipid profile outcome in a forest plot.

## 4. Discussion

Psoriasis, a chronic, genetic, immune-mediated skin disease affecting both genders equally, impacts around 125 million people globally [38,41]. Its prevalence increases linearly from 0.12% at 1 year of age to 1.2% by 18 [42]. The autoimmune response primarily involves the IL-23-mediated activation of the Th17 pathway, leading to downstream events like keratinocyte proliferation and immune cell infiltration [43]. Factors such as genetic predisposition, aging, climate, sun exposure, and ethnicity contribute to psoriasis susceptibility [44]. The disease presents various subtypes, including plaque, guttate, eruptive, inverse, pustular, and erythrodermic psoriasis, with plaque psoriasis being the most common (90% of cases) [45]. Due to the association between psoriasis and accelerated atherosclerosis, lipid-lowering drugs such as statins play an important role in the management of psoriasis [46]. Statins not only have a lipid-lowering effect by inhibition of 2-HMG-CoA reductase, but it was also found that they have immunomodulatory and anti-inflammatory effects. They reduce C-reactive protein levels and inhibit tumor necrosis factor-alpha and interleukin-1 and -6, which have an important role in the pathogenesis of psoriasis. They also inhibit natural killer cell activity and LFA-1. All of this was found to reduce the PASI score [47].

A systematic review that was carried out in 2016 included only three clinical trials assessing the efficacy of statins on the severity of psoriasis and concluded that the findings demonstrated inconsistent outcomes; nevertheless, within the excluded studies, predominantly involving single-arm, non-placebo-controlled designs, a majority indicated an enhancement in PASI scores following statin administration [24]. Furthermore, another systematic review on lipid-lowering agents, including RCTs, single-arm studies, and in vitro studies, concluded that cholesterol-reducing medications might alleviate symptoms in psoriasis patients. Notably, statins, a type of lipid-lowering drug, have shown promising efficacy and minimal side effects in treating psoriasis [25]. However, neither of the two studies performed a meta-analysis of pooled effect estimates, making this paper the first meta-analysis to evaluate pooled results of statins for psoriasis.

### 4.1. Main Findings

In this systematic review, the primary outcomes focused on PASI and DLQI. The meta-analysis for PASI scores at week 4 showed no statistically significant difference between statins and placebo. In contrast, at week 8, there was a significant reduction in PASI score for the statin group compared to placebo. However, at week 12, no significant difference was observed. The DLQI results showed initial heterogeneity, which was resolved through sensitivity analysis. The secondary outcomes included a 75% reduction in PASI and lipid profile changes. The previous analysis revealed no significant difference between the groups, while the other revealed a significant reduction in cholesterol and triglyceride levels for the statin group, with some limitations due to heterogeneity and the weight of results from a single study. Overall, the findings provide insights into the efficacy of statins in managing psoriasis, but further research may be needed to address heterogeneity and improve the robustness of the results.

### 4.2. Implications

The clinical implications of this systematic review and meta-analysis on psoriasis are significant, suggesting that statins may offer a beneficial adjunctive therapy in the management of this chronic autoimmune condition. The findings indicate that statins can improve the PASI scores, particularly noticeable at the 8-week mark, and enhance the DLQI, which underscores improvements in patients’ quality of life. The immunomodulatory and anti-inflammatory effects of statins, along with their lipid-lowering properties, contribute to these outcomes. This dual benefit is particularly crucial given the association between psoriasis and increased cardiovascular risk.

### 4.3. Limitations

This study has some limitations: (I) The currently available number of clinical trials testing the effectiveness and safety of statins for patients with psoriasis is limited, and more sophisticated RCTs are needed to obtain conclusive evidence. (II) The results of the meta-analysis of some outcomes showed heterogeneity that could be solved but not explained.

### 4.4. Recommendations

Future studies are recommended to conduct RCTs with a larger number of participants. Additionally, protocol guidelines concerning the most effective doses for statins in psoriasis patients are yet to be established. Despite the fact that results for 6 months’ follow-up have been presented, some concerns might arise as no pooled analysis was performed. In this review, a pooled analysis was conducted during the first 12 weeks only. Considering the chronicity of psoriasis, it is crucial to highlight the importance of assessing the treatment’s sustained efficacy beyond 12 weeks. Therefore, studies over a longer timeframe are required to determine whether the effectiveness of statin therapies remains consistent or shows further improvement over time. In addition, it is essential to conduct head-to-head studies against active comparators such as biological therapies, steroids, or immunosuppressive drugs to compare the effectiveness and safety of statins versus existing commercially available treatments.

## 5. Conclusions

In summary, this systematic review and meta-analysis imply that statins can be beneficial for managing psoriasis, particularly in terms of PASI scores and DLQI. Statins showed significant improvements in PASI scores and DLQI at week 8, suggesting potential benefits for patients’ disease severity and quality of life. However, further research is needed to determine the optimal treatment duration and address the observed heterogeneity in some outcomes. The secondary findings reveal no significant difference in PASI 75% reduction rates and highlight the potential additional benefits of statins in reducing cholesterol and triglyceride levels. In conclusion, these results provide preliminary evidence for the potential of statins in psoriasis management, but more studies are required to establish their role and optimize their application in clinical practice.

## Figures and Tables

**Figure 1 healthcare-12-01526-f001:**
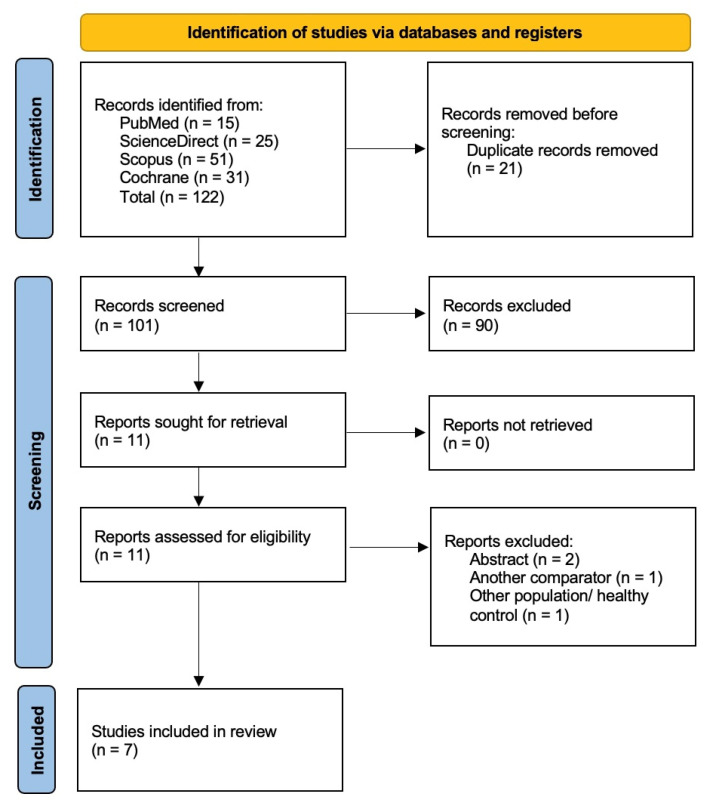
The PRISMA flow diagram of the selection process.

**Figure 2 healthcare-12-01526-f002:**
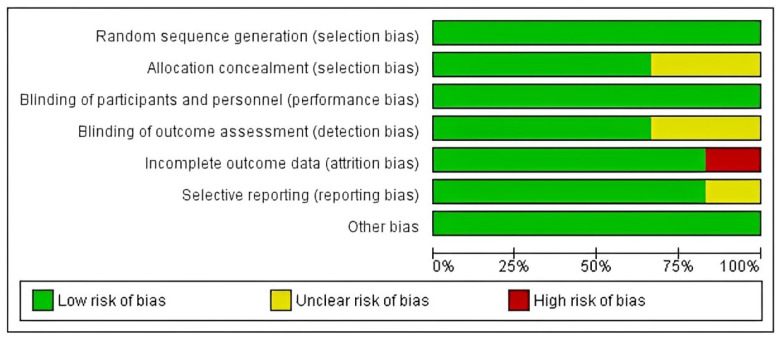
Risk of bias graph.

**Figure 3 healthcare-12-01526-f003:**
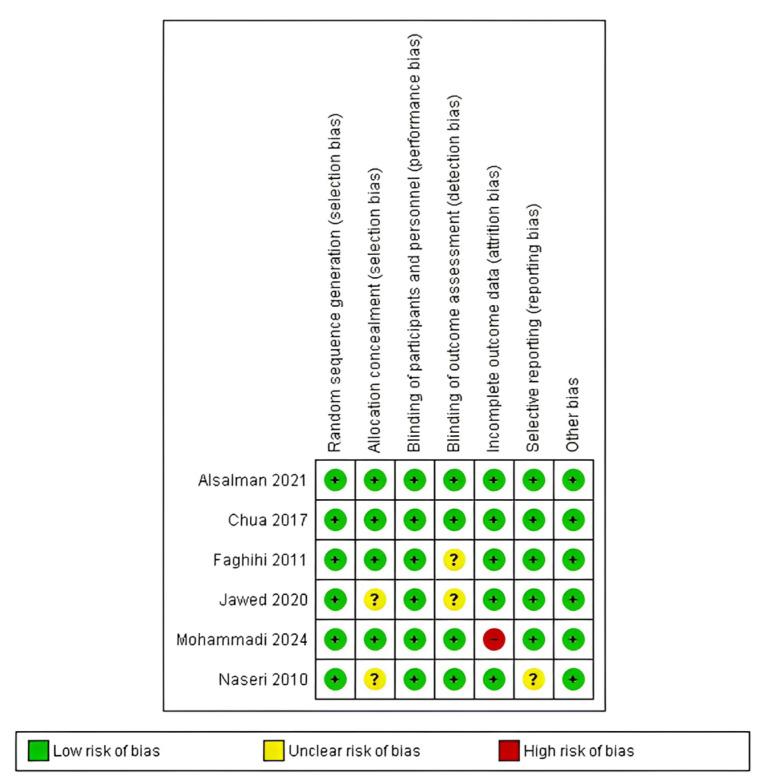
Risk of bias summary [35,36,37,38,39,40].

**Figure 4 healthcare-12-01526-f004:**
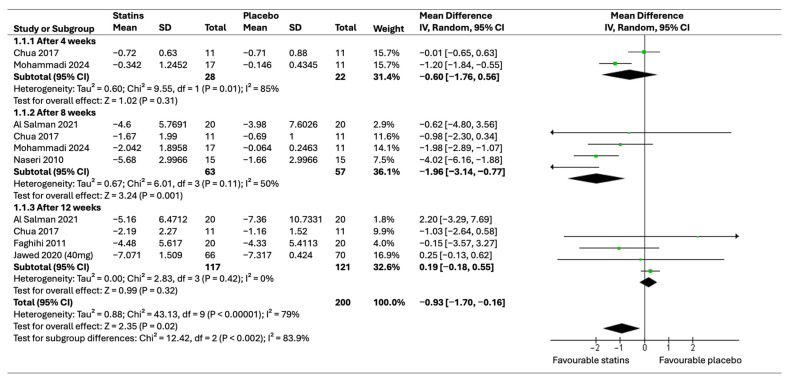
Forest plot of pooled effect of PASI score at 4, 8, and 12 weeks [35,36,37,38,39,40].

**Figure 5 healthcare-12-01526-f005:**
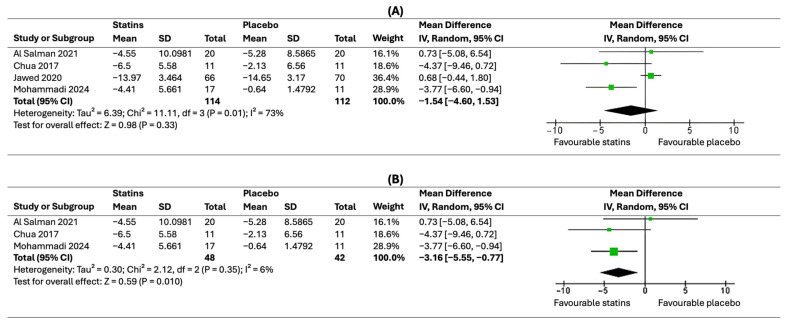
Forest plot of meta-analysis of DLQI score [35,36,38,39]: (**A**) Sensitivity analysis, before excluding Jawed et al. [38]; (**B**) after the exclusion of Jawed et al. [38].

**Figure 6 healthcare-12-01526-f006:**
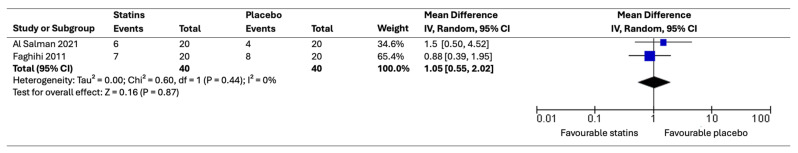
Forest plot of meta-analysis of patients with PASI 75% reduction [35,37].

**Figure 7 healthcare-12-01526-f007:**
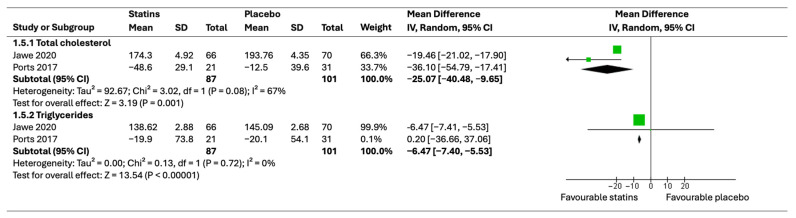
Forest plot of lipid profile outcomes [38,41].

**Table 1 healthcare-12-01526-t001:** Summary of included studies.

Study	Study Type	Location	Study Duration (Month)	Population	Interventions	Control	Sample Size	
							Intervention	Control
Al Salman et al. [35], 2021	RCT	Iran	3	Patients with moderate to severe plaque-type psoriasis (PASI score under 20) undergoing NB-UVB phototherapy.	Simvastatin 40 mg/day + NB-UVB	Placebo + narrow-band UVB	22 (2 lost)	22 (2 lost)
Faghihi et al. [37] 2011	RCT	Iran	3	Patients aged 16 to 60 years diagnosed with acute or chronic plaque-type psoriasis, with body surface area (BSA) involvement exceeding 10%.	Atorvastatin 40 mg/day	Placebo	20	20
Jawed et al. [38], 2020	RCT	Pakistan	6	Patients of both genders aged 25 to 65 years with psoriasis, having a PASI score of less than 12 and hsCRP levels of 3 or higher	G1: atorvastatin 40 mg/day for 3 months then 20 mg/day for the next 3 months G2: atorvastatin 80 mg/day for 3 months then 40 mg/day for the next 3 months **	Placebo **	G1: 75 (9 lost) G2: 75 (5 lost)	75 (5 lost)
Naseri et al. [40], 2010	RCT	Iran	2	Patients with plaque psoriasis	Simvastatin 40 mg/day **	Placebo **	15	15
Mohammadi et al. [39], 2024	RCT	Iran	2	Adult patients (18 years and older) with mild to moderate plaque psoriasis	G1: melatonin 0.5% cream twice/d G2: Topical Rosuvastatin 0.5% twice per day for 8 weeks	Placebo	G1: 27 (3 lost) G2: 25 (8 lost)	25 (14 lost)
Chua et al. [36], 2017	RCT	Philippines	6	Patients between the ages of 19 and 65 diagnosed with mild to moderate chronic plaque psoriasis, having PASI scores below 10	Atorvastatin 40 mg/day **	Placebo **	14 (3 lost)	14 (3 lost)
Ports et al. [41], 2017	post hoc analysis	United Kingdom		Patients with history of psoriasis or receiving its medications involved in three different trials	Atorvastatin 10 mg/day	Placebo	21	31

PASI: Psoriasis Area Severity Index; RCT: randomized clinical trial, NB-UVB: narrow-band ultraviolet rays type B; HsCRP: high-sensitivity C-reactive protein. ** both groups received additional topical steroids.

**Table 2 healthcare-12-01526-t002:** Characteristics of patients in the included studies.

Study	Study Arms	Sample Size	Age Mean ± SD	Sex, Males n (%)	Smoking (Yes) N (%)	PASI Mean (±SD)	DLQI Mean (±SD)
Al Salman et al. [35], 2021	Simvastatin	20	41 (±18.08)	10 (50%)	NR	9.34 (±3.75)	10.2 (±8.13)
	Placebo	20	46.9 (±17.4)	12 (60%)	NR	11.715 (±9.02)	9.4 (±6.85)
Faghihi et al. [37] 2011	Atorvastatin	20	43.85 (±14.3)	8 (40%)	NR	7.42 (±1.9)	NR
	Placebo	20	36.55 (±12.1)	12 (60%)	NR	6.92 (±1.76)	NR
Jawed et al. [38], 2020	Atorvastatin 40 mg/day	70	47.8 (±8.23)	54 (77.15)	44 (62.86%)	10.89 (±1.19)	19.6 (±1.98)
	Atorvastatin 80 mg/day	67	47.9 (±8.42)	52 (76.48)	43 (63.24)	10.99 (±0.711)	19.77 (±1.43)
	Placebo	66	46.5 (±7.84)	57 (86.37%)	42 (63.64%)	11.23 (±0.729)	20.2 (±1.16)
Naseri et al. [40], 2010	Simvastatin	15	38.5 (±13.8)	9 (60%)	NR	9.51	NR
	Placebo	15	45.4 (±15.52)	11 (73.3%)	NR	5.64	NR
Mohammadi et al. [39], 2024	Rosuvastatin	17	42.94 (±9.97)	7 (41.2%)	3 (17.6%)	2.91 (±1.85)	12.53 (±8.43)
	Placebo	11	36.55 (±9.12)	4 (36.4%)	1 (9.1%)	1.76 (±1.23)	8.64 (±4.2)
Chua et al. [36], 2017	Atorvastatin	14	41.29 (±11.38)	7 (50%)	NR	5.49 (±2.78)	11.5 (±6.04)
	Placebo	14	40.71 (±12)	4 (28.57%)	NR	5.63 (±2.52)	9.07 (±5.84)

PASI: Psoriasis Area Severity Index. DLQI: Dermatology Life Quality Index. NR: not reported.

## Data Availability

All data are publicly available.

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
