# Peer review of "The Impact of Statins on Disease Severity and Quality of Life in Patients with Psoriasis: A Systematic Review and Meta-Analysis"

_healthcare, 2024, doi:10.3390/healthcare12151526_

Round 1

Reviewer 1 Report

Comments and Suggestions for Authors

Thank-you for a very robust meta-analysis. I found your methodology sound and your findings appropriate. I would suggest however that the results be clearly stated as part of the abstract as it wasn't until the end of the paper that the 12 week testing process revealed that there was no significant improvement, even though there was 8 weeks. This in and of itself is an intersting finding to recommend exploration. 

Author Response

Dear Reviewer 1,

Comment 1: Thank-you for a very robust meta-analysis. I found your methodology sound and your findings appropriate. I would suggest however that the results be clearly stated as part of the abstract as it wasn't until the end of the paper that the 12 week testing process revealed that there was no significant improvement, even though there was 8 weeks. This in and of itself is an interesting finding to recommend exploration.

Response 1: Thank you for your insightful feedback. Revised, please see Page 1 Line 35 - 38. 

Reviewer 2 Report

Comments and Suggestions for Authors

Dear authors,

Congratulations on your article. The information provided is exciting.

I recommend revising the following paragraphs because the percentage match shown in the plagiarism report shows a similarity of 23%, which I consider too high for the article.

2.4. Primary outcomes- paragraph 1

2.6. Risk of bias assessment

2.7. Statistical analysis

3.4.1 PASI

- My main issue regarding the manuscript is the high similarity rate within the paragraphs I mentioned, which is close to 23%, percentage I consider worrisome. More specifically, the following paragraphs were highlighted:

“The primary outcomes of this study were psoriasis severity and QoL. Psoriasis sever- 146 ity was investigated using the Psoriasis Area And Severity Index (PASI) after 4 weeks, 8 147 weeks and 12 weeks reflecting the extent of psoriasis and its severity in which the body is 148 divided into four regions for assessment: head (h), upper extremities (u), trunk (t), and 149 lower extremities (l), which collectively account for 10%, 20%, 30%, and 40% of the total 150 body surface area (BSA), respectively. Each region is evaluated separately for erythema, 151 induration, and scaling, rated on a scale from 0 (none) to 4 (very severe). The extent of 152 psoriatic involvement was categorized as follows: 0 (No involvement); 1 (1% to 9%); 2 153 (10% to 29%); 3 (30% to 49%); 4 (50% to 69%); 5 (70% to 89%); 6 (90% to 100%). The PASI 154 score were calculated using the formula: 155 PASI = 0.1 (Eh + Ih + Sh) Ah + 0.2 (Eu + lu + Su) Au + 0.3 (Et +lt + St) At + 0.4 (El +ll 156 +Sl) Al, For quality of life, it was assessed using the Dermatology Life Quality Index (DLQI), 161 which is a 10-item questionnaire designed to assess six different aspects that may impact 162 quality of life, including symptoms and feelings, leisure, daily activities, personal rela- 163 tionships, work and school performance, and treatment. Each aspect of the assessment 164 has a maximum score of either 3 (from a single question) or 6 (from two questions). The 165 scores for these aspects were expressed as a percentage of their respective maximum val- 166 ues. Each of the 10 questions is rated on a scale from 0 (not at all) to 3 (very much). The 167 overall DLQI is determined by summing the scores of all questions, resulting in a numer- 168 ical score ranging from 0 to 30 or a percentage of 30. Higher scores indicate a greater im- 169 pact on the patient's quality of life. The DLQI scores can be interpreted as follows: No 170 effect (0 to 1); Small effect (2 to 5); Moderate effect (6 to 10); Very large effect (11 to 20); 171 Extremely large effect (21 to 30).”

“The risk of bias for the included RCTs was conducted according to the Cochrane Risk 178 of Bias tool (ROB1) of interventional studies stated in their book, which encompasses se- 179 lection bias and allocation concealment, blinding of patients and personnel, Blinding of 180 outcome assessors, missing outcomes data, selective reporting of outcomes and other 181 sources of bias if present. Two authors independently rated each domain as a high, low, 182 or unclear risk.”

“The meta-analysis was performed with the inclusion of at least two studies with 185 available data for assessed outcomes using RevMan software version 5.4.1 [26]. Regarding 186 continuous outcomes, data was pooled as mean difference (MD) with a 95% confidence 187 interval (CI). For dichotomous outcome data, the events frequency and the total number 188 of patients were pooled as risk ratio (RR) with a 95% CI. The level of statistical significance 189 was set to be p < 0.05. The change from baseline values was extracted instead of final 190 values. In case of the absence of the SD of change, an estimate was computed using the 191 reported p-values. A random effect model (inverse variance) was adopted rather than a 192 fixed effect model, yielding a more conservative estimate of the pooled effect and gener- 193 alizable results. To evaluate the presence and degree of heterogeneity, Chi-square and I- 194 Healthcare 2024, 12, x FOR PEER REVIEW 5 of 16 square tests were used, respectively, as outlined in chapter nine of the Cochrane Hand- 195 book. Interpretation of I-square test results was carried out as follows: 0-40% were con- 196 sidered insignificant, 30-60% were considered moderate, and more than 50% were consid- 197 ered substantial. Heterogeneity was considered significant if the alpha level for the Chi- 198 square test was below 0.1. Finally, we could not assess publication bias using funnel plots 199 due to the limited number of included studies.”

“In a comprehensive meta-analysis of a pooled effect of clinical trials regarding this 250 outcome, PASI score was assessed at 4 weeks, 8 weeks and 12 weeks after initiation of 251 treatment. PASI score at week 4 was reported in two studies Chua et al., and Mohammadi 252 et al. Results showed no statistically significant difference between statins and placebo, 253 pooled effect (MD= -0.60, 95%CI [-1.76,0.56]) with a p-value of 0.31. Data showed signifi- 254 cant heterogeneity (p=0.01 and I2=85%). PASI score at week 8 was stated on four in- 255 cluded trials [27], [31], [32]. The results of the meta-analysis indicate a statistically signifi- 256 cant reduction in the PASI score in the Statin group compared to the placebo (MD=-1.96, 257 95%CI [-3.14,-0.77]) indicated by a p-value of 0.001. Results showed no marked heteroge- 258 neity was detected (p=0.11 and I2=50%). Moreover, at week 12, results were reported on 259 four studies [27, 28, 29, 30]. The results also revealed that there is no statistically significant 260 difference between statins and placebo at week 12 (MD=0.19, 95%CI [-0.18,0.55]) indicated 261 by a p-value of 0.32 data showed no heterogeneity (p=0.42, I2=79%). Figure 4 shows a 262 forest plot of the meta-analysis of the PASI score.”

Thus, I recommend revising and rephrasing these areas.

- Regarding methods, I would like more details regarding the  Risk of Bias Assessment (what happened if the authors disagreed, how did they reach a middle ground if so) and if there is any possibility, of preferred statins used and if there was at some point a difference in the response/PASI score of the patients.

- The bibliography and other aspects are good.

Author Response

Dear Reviewer 2,

Comment 1: I recommend revising the following paragraphs because the percentage match shown in the plagiarism report shows a similarity of 23%, which I consider too high for the article.

Response 1:  We greatly appritiate invaluable comments and insightful feedback. The manuscript was revised accordingly for all the specified paragraphs.

Comment 2: - Regarding methods, I would like more details regarding the  Risk of Bias Assessment (what happened if the authors disagreed, how did they reach a middle ground if so) and if there is any possibility, of preferred statins used and if there was at some point a difference in the response/PASI score of the patients.

Response 2: Revised, please see Page 4: Risk of Bias Assessment

Reviewer 3 Report

Comments and Suggestions for Authors

Dear Authors,

I found your manuscript very interesting. Anyhow, I have some comments.

The manuscript is well written, and the results properly described. But I think the discussion section could be developed and expanded. Now there is a duplicate of the introduction in the discussion which is not necessary since the information is found in the beginning of the paper and furthermore the text does not support the results. There could be more of discussion about why the results seemed to be a little week. What other limitations could be of interest, and could there be other explanations such as other effects of the treatment? What could improve studies on this subject? 

Author Response

Dear Reviewer 3,

Thank you for your invaluable comments and insightful feedback.

Comment 1: Now there is a duplicate of the introduction in the discussion which is not necessary since the information is found in the beginning of the paper and furthermore the text does not support the results. 

Response: The discussion section was revised and replicated information (found in the introduction) were deleted (deleted paragraph:

 Treatment approaches range from corticosteroids, vitamin D analogues, and calcineurin inhibitors for mild or localized cases to targeted phototherapy, systemic treatments like biologics and oral agents, and topical treatments for moderate to severe psoriasis. Biologics, targeting specific cytokine molecules in the inflammatory cascade, ¬of cardiovascular risk).

Comment 2: What other limitations could be of interest, and could there be other explanations such as other effects of the treatment? What could improve studies on this subject? 

Response: The manuscript was revised, please see Page 14; 

     Future studies are recommended to conduct RCTs with a larger number of participants. Additionally, protocol guidelines concerning the most effective doses for statins in psoriasis patients are yet to be established. Despite that the results for 6 months follow-up is present, some concerns might arise as no pooled analysis was performed. In this review, pooled analysis was conducted during the first 12 weeks only. Considering the chronicity of psoriasis, it is crucial to highlight the importance of assessing the treatment's sustained efficacy beyond 12 weeks. Therefore, studies over a longer timeframe are required to determine whether the effectiveness of statin therapies remains consistent or shows further improvement over time. In addition, it is essential to conduct head-to-head studies against active comparators such as biological therapies, steroids, or immunosuppressive drugs to compare the effectiveness and safety of statins versus existing commercially available treatments.

Reviewer 4 Report

Comments and Suggestions for Authors

Overall, this article provides valuable insights into the potential role of statins in the treatment of psoriasis through a systematic review and meta-analysis. However, there is room for improvement, particularly in the methods, results, and discussion sections. It is hoped that the authors can revise the article according to the following suggestions to enhance its quality.

 Introduction

1. The introduction of statins is not detailed enough, especially regarding their application in other immune-mediated diseases.

2. The relationship between psoriasis and cardiovascular diseases is not discussed, nor is the dual role of statins in managing these conditions.

 Methods

3. The search terms and logical combinations are not specific enough.

4. The standards for diagnosing psoriasis and the characteristics of patients are not described in detail.

5. The data extraction section lacks an explanation of how incomplete data and inconsistencies between studies were handled.

6. The bias risk assessment section does not provide detailed results and processes.

 Results

7. There is a lack of in-depth analysis and discussion on the causes and impacts of heterogeneity.

8. Subgroup analysis is missing for different types and doses of statins.

9. The process and results of the sensitivity analysis are not detailed enough.

10. The explanations accompanying the figures and tables are insufficient.

11. It is recommended to improve the image quality and clarity of Figures 3 to 6.

 Discussion

12. There is no comparison with other related studies, and the innovative aspects and contributions of this study are not highlighted.

13. The discussion on the specific application scenarios and potential challenges of using statins in treating psoriasis is not detailed enough.

14. The discussion on the study's limitations is not comprehensive, such as the small sample size and the regional limitations of the included studies.

Author Response

Dear Reviewer 4,

Thank you for your invaluable comments and insightful feedback.

Introduction

1. The introduction of statins is not detailed enough, especially regarding their application in other immune-mediated diseases.

2. The relationship between psoriasis and cardiovascular diseases is not discussed, nor is the dual role of statins in managing these conditions.

Response: The manuscript was revised accordingly, please see Page 2 Line 90-100 &  Page 3 Line 101-104

 Methods

3. The search terms and logical combinations are not specific enough.

Response: Revised; please see Page 3; Database Searching

4. The standards for diagnosing psoriasis and the characteristics of patients are not described in detail.

Response: Revised, please see Page 3;  Inclusion and Exclusion Criteria

5. The data extraction section lacks an explanation of how incomplete data and inconsistencies between studies were handled.

Response: Revised, please see Page 4; Study Selection and Data Extraction

6. The bias risk assessment section does not provide detailed results and processes.

Response: Revised, please see Page 4 & 5; Risk of Bias Assessment

Results

7. There is a lack of in-depth analysis and discussion on the causes and impacts of heterogeneity.

Response: Revised; please see Page 11, Line 278 to 281, Page 12 Line 298 to 300, Page 14 Line 363 to 369.

8. Subgroup analysis is missing for different types and doses of statins.

9. The process and results of the sensitivity analysis are not detailed enough.

Response: Please see P11 & 12 

10. The explanations accompanying the figures and tables are insufficient.

Response: Revised

11. It is recommended to improve the image quality and clarity of Figures 3 to 6.

Response: Thank you for your constructive advice, however, this is the highest resolution available at the moment.

Discussion

12. There is no comparison with other related studies, and the innovative aspects and contributions of this study are not highlighted.

Response: Please see Page 13, second paragraph & implications.

13. The discussion on the specific application scenarios and potential challenges of using statins in treating psoriasis is not detailed enough.

Response: Revised, please see Page 14 Implications.

14. The discussion on the study's limitations is not comprehensive, such as the small sample size and the regional limitations of the included studies.

Response: Revised, details regarding the limitations were addressed for future studies, please see Page 14; Recommendations.

If any further concerns or needed revisions please let us know, thank you.

Round 2

Reviewer 2 Report

Comments and Suggestions for Authors

Thank you for providing the requested changes. From my point of view, no further modifications are required.

Author Response

Thank you, I really appreciate your time and effort in reviewing my work.

Reviewer 4 Report

Comments and Suggestions for Authors

The authors responded to almost all my concerns, and the quality of the revised manuscript has been greatly improved.

Author Response

(The authors gave the same response as above.)
